# Hereditary Hemorrhagic Telangiectasia: Genetics, Pathophysiology, Diagnosis, and Management

**DOI:** 10.3390/jcm11175245

**Published:** 2022-09-05

**Authors:** Adrian Viteri-Noël, Andrés González-García, José Luis Patier, Martin Fabregate, Nuria Bara-Ledesma, Mónica López-Rodríguez, Vicente Gómez del Olmo, Luis Manzano

**Affiliations:** 1Internal Medicine Department, Hospital Universitario Ramón y Cajal, IRYCIS, 28034 Madrid, Spain; 2Faculty of Medicine and Health Sciences, Universidad de Alcalá (UAH), 28801 Alcalá de Henares, Spain

**Keywords:** hereditary hemorrhagic telangiectasias, arteriovenous malformations, angiogenesis, epistaxis, telangiectasias, VEGF

## Abstract

Hereditary hemorrhagic telangiectasia is an inherited disease related to an alteration in angiogenesis, manifesting as cutaneous telangiectasias and epistaxis. As complications, it presents vascular malformations in organs such as the lung, liver, digestive tract, and brain. Currently, diagnosis can be made using the Curaçao criteria or by identifying the affected gene. In recent years, there has been an advance in the understanding of the pathophysiology of the disease, which has allowed the use of new therapeutic strategies to improve the quality of life of patients. This article reviews some of the main and most current evidence on the pathophysiology, clinical manifestations, diagnostic approach, screening for complications, and therapeutic options, both pharmacological and surgical.

## 1. Introduction

### History and Epidemiology

Hereditary hemorrhagic telangiectasia (HHT), also known as Rendu–Osler–Weber Syndrome, is a rare disease characterized by multisystemic vascular dysplasia [1].

Although it was first described by the British pathologist Henry Gawen Sutton in 1864 [2], it would be another 32 years before it was first differentiated from hemophilia by the French physician Henri Jules Louis Marie Rendu [3].

This disease can present a wide variability of clinical manifestations and can differ markedly even among members of the same family. Its prevalence is, according to recent estimates, 1 case per 5000 inhabitants [4]. However, there are geographical areas where frequencies are higher, such as the Dutch Antilles, the island of Funen, Denmark, the French region of Ain, Vermont (USA), Newcastle (United Kingdom), and Las Palmas de Gran Canaria (Spain) [5].

This disease is caused, in approximately 90% of cases, by a heterozygous mutation of the endoglin gene (*ENG*) or the activin-like receptor kinase 1, called ALK1 (also known as *ACVRL1*), characterized by an autosomal dominant inheritance pattern. In addition to these two genes, alteration of SMAD4 has been identified in a subset of patients with HHT and juvenile polyposis (approximately 2% of cases), a condition termed PJ-HHT syndrome, in which juvenile polyps and anemia are the predominant clinical features. Disruption of the *ENG* gene, located on chromosome 9 (9q3.3-q3.4), causes HHT type 1 (HHT1), while mutation of the *ACVRL1* gene, located on chromosome 12 (12q13), causes HHT type 2 (HHT2) [5]. Table 1 shows the main genes responsible for HHT, including all the phenotypes described to date [6,7]. Other less frequently affected genes have been described such as *GDF2* and *RASA-1*. There is a close overlap between capillary malformation–arteriovenous malformation (CM–AVM) syndrome and HHT, both related to the *RASA-1* gene mutation but expressing different phenotypes [7,8,9,10].

## 2. Pathophysiology

### 2.1. Regulation of Angiogenesis

The relationship between the endothelium and affected genes in HHT is based on the positive regulation of angiogenesis by *ENG* and *ACVRL1* [11], a process in which the pathways of endothelial cell migration and proliferation are central. All affected genes encode proteins that participate in the transforming growth factor β (TGF-β) signaling pathway [4,11]. The TGF-β family is a large and continuously expanding group of regulatory polypeptides. They are subdivided into two functional groups, the first includes the activins, inhibins, and the nodal growth differentiation factor; and the second includes the bone morphogenetic proteins (BMPs), most of the growth differentiation factors (GDFs), and antimullerian hormone (AMH), among others [12].

BMP 9 and 10 are the main ligands of the TGF-β family involved in this HHT pathway. They exert their action through a heteromeric serine/threonine type I (RI) and type II (RII) transmembrane receptor and then through the Smad cascades. Two types of RI receptors are involved in HHT, ALK-1 and ALK-5, with different signaling Smad cascades (Smad1/5/8 and Smad2/3, respectively). Endoglin plays a role as auxiliary receptor for dimerization and enhanced activation of ALK-1 and RII receptors. A soluble form can be generated by proteolysis of the membrane-bound receptor that can sequester ligands (BMP9 or 10) and thus modulate their binding to R-I/R-II receptors. The Smad family complexes translocate to the nucleus and contribute to the transcription of genes involved in matrix development or migration of the endothelium [11,13]. ALK1 and ALK5 generate different signals and can play opposite roles in angiogenesis. TGF-β/ALK1 induces endothelial cell migration and proliferation, while TGF-β/ALK5 inhibits these effects and promotes extracellular matrix deposition. Depending on the circulating levels of TGF-β family members, one pathway is going to predominate compared to the other. In a low TGF-β environment, ALK1/Smad1/5/8 route will enhance cell proliferation and migration (active phase), while in a high TGF-β environment, ALK5/Smad2/3 route will promote extracellular matrix deposition (quiescent phase) [14,15,16]. These pathways potentially involved in the pathophysiology of HHT are depicted in Figure 1.

### 2.2. Pathogenesis in HHT

The haploinsufficiency model in HHT is the most accepted theory for the pathogenesis of disease development [17]. It is explained by the fact that mutations in *ENG* and *ACVRL1* genes generate altered proteins that fail to be expressed in the cell membranes of endothelial cells. There have been more than 900 mutations described on the *ENG* and *ACVRL1* genes including deletions, insertions, nonsense, missense, and splice site mutations [18]. This lack of protein expression results in impairment on routes via *ENG* or *ACVRL1* that normally enhances endothelial migration and proliferation. Studies have shown diminished cell surface expression of endoglin and ALK1 in HHT patients [15,18]. Importantly, animal models show that involvement of the endoglin and ALK1 pathways will also lead to a decrease in ALK5 pathway activity as a possible adaptation response to compensate the reduced expression of endoglin and ALK1 [19,20]. This would explain why there is an alteration in both the migration of endothelial cells and the subsequent formation of the cellular matrix in this pathology [17].

Two other less recognized theories about the development of HHT are the dominant negativity theory and the double knockout hypothesis. The first searches to explain the development of the disease in patients with nonsense mutations in the *ENG* and *ACVRL1* genes. These truncated proteins could inhibit endoglin protein expression on the vascular endothelium [8,21]. On the other hand, somatic mutations in telangiectases of the skin [22] have driven to propose a double hit hypotheses. In this scenario, more localized lesions (such as cutaneous telangiectasias) would be explained by somatic mutations that are caused by environmental stress (sun exposure). This hypotheses could help to explain the increase in number of manifestation with the age of the patients [21,22]. Familial case series have shown related germline mutations and worse prognostic phenotypes linked to advanced age. This suggests that inflammatory processes over time may produce additional somatic mutations that influence the phenotype of patients [23,24]. Furthermore, patients with a mutation in both genes have been described in the literature without necessarily presenting a more pathological phenotype [25]. This exemplifies the need to look for more theories to explain the pathophysiology of the disease.

### 2.3. Relationship between Inflammation and Hereditary Hemorrhagic Telangiectasia

Endoglin could help as an adhesion molecule for leukocyte infiltration [26]. It has been proposed that during inflammation, endoglin is excised and its soluble fraction (sol-eng) could promote adhesion and chemotaxis for mononuclear cells (MNC) [26]. In HHT, leukocyte infiltration mediated by the expression of adhesion molecules and chemokines synthesized by the endothelium could be impaired, affecting vascular repair and remodeling [11,27].

Endoglin and ALK1 are not only expressed in endothelial cells but are also found in the MNCs. They are involved on maturation on the bone marrow and migration into the circulation of the MNCs [28]. The alteration in signaling via endoglin and ALK1 in MNCs has been proposed as a justification for the alteration in the immune response present in these patients, such as a higher incidence of infections or leukopenia [29,30]. The existence of innate cellular immunodeficiency mediated by macrophages has also been proposed, having demonstrated alterations in the migration and release of mediators or interleukins in patients with HHT [30]. Indeed, haploinsufficiency in *ENG* is related to a decrease in lymphopoiesis and poor activation of B cells to produce antibodies [28]. In this sense, it has been shown that *ENG* knockout mice presented an increase in spontaneous infections and a lower inflammatory response [30].

### 2.4. Relationship between Hemostasis and Hereditary Hemorrhagic Telangiectasia

Another approach currently being developed attempts to identify alterations in the coagulation and fibrinolysis cascades in HHT [17]. It has been identified that patients with HHT present elevated levels of factor VIII and von Willebrand factor, indicating that these patients could present a higher risk of venous thrombosis than the general population [31,32,33]. On the other hand, it has been shown that sol-eng mediates platelet adhesion through the IIb3 integrin complex present in platelets [34]. This could explain why HHT patients, in addition to having an increased risk of bleeding, also tend to bleed longer and more profusely.

## 3. Clinical Manifestations

### Clinical Presentation

Table 2 shows the different clinical manifestations and complications that occur in patients with HHT, as well as their approximate prevalence. Studies based on the Spanish registry (Registro informatizado de la telangiectasia hemorrágica hereditaria, RiTHHA, in spanish) showed a different correlation between genotype and phenotypic expressions. Both groups had similar frequencies in terms of the frequency of epistaxis, telangiectasias, and anemia, but in terms of AVMs there were differences between the two groups [35]. Although the clinical phenotype of both pathologies is similar, HHT2 has a lower penetrance and a later onset of symptoms than HHT1.

Recurrent, spontaneous epistaxis (nosebleeds) occurs in approximately 50% of patients by 20 years and go up to 90% by the age of 40. They are the main cause of functional impairment in these patients since they lead to iron deficiency and anemia [36]. The severity of this symptom is measured using the ESS (Epistaxis Severity Score) scale, which includes six questions on the frequency, intensity, or need for medical attention of bleeding [37].

Telangiectases are lesions formed from malformation of small arterioles and venules within the capillary beds and expresses as pinpoint to pinched-sized lesions. They have been described primarily on the lips, tongue, face, fingers, and in the gastrointestinal tract, and oral and nasal mucosa. They are more prevalent on adults as they appear with age with only 30% appearing before the age of 20 years. As they are very fragile, telangiectasias are prone to bleed and limit the quality of living of patients [38]. 

Iron deficiency anemia is a very prevalent finding given recurrent nosebleeds. The mean age of onset of these is estimated to be around 12 years and they increase with age. Anemia and epistaxis are the principal symptoms that impact the quality of life, even causing dyspnea and palpitations depending on their severity. Maintaining hemoglobin levels above 8–9 g/dL by red blood cell transfusions can be considered to avoid low oxygen supply to peripheral tissues in patients with very symptomatic HHT (chronic uncontrolled bleeding), comorbidities (heart failure, hypoxaemia from pulmonary AVM), and urgent situations (such as surgery or pregnancy) [36]. On clinically stable patients with no cardiac or pulmonary comorbidities, oral iron supplementation is indicated to maintain a hemoglobin level above 7 g/dL and a ferritin level above 100 µg/L.

## 4. Vascular Malformations and Its Complications

### 4.1. Pulmonary Arteriovenous Malformations (PAVMs)

They are present in approximately 50% of patients with HHT with a predominance in HHT1 [23,34]. They are divided into degrees of severity, the most frequent are a simple connection between a segmental pulmonary artery and a pulmonary vein forming an aneurysmal niche, the most complex are formed by multiple arteries draining into multiple veins and less frequently can become diffuse located within an entire pulmonary segment. The latter are those with the highest risk of embolism and hypoxaemia. Figure 2 illustrates a patient with a pulmonary AVM. Most PAVMs are asymptomatic but can rarely cause hypoxaemia and/or digital clubbing. Other rare complications include hemoptysis or hemothorax. Another complication derived from these malformations is the risk of aseptic and septic embolisms that increase the risk of transient ischemic attacks, stroke, and cerebral abscesses in these patients [35]. Multiple PAVMs have been associated with an increased risk of stroke, while male sex and hypoxaemia appear to increase the risk of brain abscess [35,37,38].

### 4.2. Hepatic Vascular Malformations

The prevalence of these malformations fluctuates between 41–74% [36] and are classified anatomically in shunts between hepatic artery and portal vein; between hepatic artery and vein; and between portal vein and hepatic vein. Given that not all these communications are “arteriovenous”, a more appropriate term for them is hepatic vascular malformations (HVM) [39,40].

Depending on their primary lesion, HVMs can present with various complications. These include high output heart failure (HOHF), portal hypertension, hepatic encephalopathy, biliary ischemia, mesenteric ischemia, and liver cirrhosis. Observational studies show that patients with alteration of the *ACVRL1* gene are generally the most affected, and that there is a female predominance of 4.5 to 1. The most frequent clinical manifestations of HVMs are symptoms related to HOHF, including dyspnea and edema. Other patients can present with abdominal pain from biliary ischemia or anicteric cholestasis [39].

### 4.3. AVMs of the Gastrointestinal System

These lesions can cause gastrointestinal bleeding with a prevalence of 30% in patients with HHT and typically occur between the fifth and sixth decade of life. AVMs are mainly found in the stomach (46–75%) and small intestine (56–91%), mostly in the duodenum; and less frequently in the colon (30%). An increase in these AVMs has been observed with age and patient genotype *ENG* [34,39].

### 4.4. AVMs of the Nervous System

There is a variety of different brain AVMs, and the risk of bleeding can vary among them. Most patients present with nonspecific neurological symptoms, such as headache or seizure, that are difficult to control, and 20% of patients present with intracranial hemorrhage [34]. Screening for brain AVMs will be discussed later in this review [41]. 

## 5. Pulmonary Hypertension and HHT

Pulmonary hypertension (PH) is defined as a mean pulmonary arterial pressure (PAP) greater than or equal to 20 mmHg at rest, measured by right heart catheterization [42]. It is classified into five main types with different pathophysiology, hemodynamic characteristics, and treatments: (I) primary pulmonary arterial hypertension (PPH), (II) PH secondary to left heart failure, (III) PH due to pulmonary disease (hypoxaemia), (IV) PH due to chronic pulmonary thromboembolism, and (V) PH of multifactorial origin [43].

Different mechanisms are used to explain the development of PH in HHT. The first requires the concomitant existence of hepatic vascular malformations. This event favors the development of HOHF in some cases, which in turn generates increased cardiac output through the pulmonary circulation leading to an increase in mean pulmonary arterial (mPA) pressure. Other contributors are chronic anemia that lead to hypoxaemia and vasoconstriction of pulmonary vasculature and the other are thromboembolic disease, seen in approximately 6% of patients with HHT [7,44].

Patients usually present symptomatically with dyspnea, palpitations, and peripheral edema. Diagnosis is made with right heart catheterization and typically there is an elevated mPA pressure, high cardiac output (CO), elevated pulmonary capillary wedge pressure (PCWP) and low pulmonary vascular resistance (PVR) [7,43].

Pulmonary arterial hypertension (PAH), which belongs to the first group, is a rare complication in HHT and is related to mutations in components of the TGF-β pathway; 75% of these mutations are found in BMPR2. However, in patients in whom this gene is not altered, it is recommended to exclude mutations in *ACVLR1*, *ENG*, *MADH4*, and *GDF2*, the first three of which are also related to HHT. Findings on the right heart catheterization differ by a normal PCWP, normal or low CO and elevated PVR along with elevated mPA pressure [7,43].

## 6. Diagnosis

### 6.1. Diagnostic Criteria

Clinically, the disease is diagnosed using the Curaçao criteria [44], which include the following four parameters: (i) spontaneous and recurrent epistaxis; (ii) mucocutaneous telangiectasias; (iii) arteriovenous visceral lesions in internal organs; (iv) familial aggregation. To define a diagnosis as “probable”, at least two criteria must be present, and to have a “definitive” diagnosis, at least three of the four criteria are needed.

At present, in addition to the clinical parameters listed in the Curaçao criteria, genetic testing is available as a diagnostic tool. In total, 90% of cases of HHT are due to a pathogenic mutation in the *ENG* or *ACVRL1* genes, although pathogenic variants related to HHT have also been described in other genes such as *MADH4*, *GDF2*, or *RASA-1*.

### 6.2. Screening and Follow up for HHT

When encountering a patient with suspected HHT in the consultation room, screening should be initiated with clinical history, physical examination, and laboratory tests. 

If the diagnosis of HHT is possible or definitive, a screening for pulmonary lesions is recommended. The recommended initial test is transthoracic contrast echocardiography (TTCE) with bubbles. Its sensitivity is between 93 and 100%, and since it is noninvasive, it is still the first step in the approach to these patients. 

The number of bubbles identified in the TTCE represents a degree of severity which is classified into three different degrees. Grade 1 shows a passage of less than 30 bubbles in the same plane. Grade 2 corresponds to a passage of between 30 and 100 bubbles. Grade 3 corresponds to a passage of more than 100 bubbles [44,45,46]. 

In patients with a negative first study or grade 1, it is recommended to follow a conservative strategy of repeating TTCE every 5 years. 

In cases obtaining a grade 2 or higher, a CT scan should be performed. In the case of a negative screening CT, follow-up with CT scans every 5 years is suggested; in the case of two negative screening CTs, it could be extended to every 10 years [47].

In patients who have undergone recent embolization, a follow-up CT scan is recommended 6 months after the procedure. Follow-up with a CT scan will be performed every 3–5 years thereafter, depending on the complexity of the patient’s AVMs.

There is recent evidence demonstrating the usefulness of TTCE for the follow-up of post-embolization patients with results comparable to CT. More information would be needed to conclude on this type of follow-up [47].

For cerebral AVMs, MRI is recommended in children and some reviews suggest this at the time of diagnosis [34,41]. Most HHT experts recommend brain imaging (one MRI with and without contrast and repeat as an adult if the first one is carried out as a child) for screening. However, there is controversial evidence regarding imaging on adults, since some studies suggest low risk of complications. Indeed, follow-up imaging tests are not recommended when the MRI, conducted in adulthood, is negative [34,41].

As for hepatic malformations, given the good sensitivity, low cost and low risk involved, the recommended test is abdominal ultrasound with Doppler [35,48,49]. If the first test is positive, additional imaging can be considered with the use of liver MRI or abdominal CT angiography to define which type of HVM is predominant [48].

Regarding digestive AVMs, screening is performed when the patient’s anemia cannot be explained by epistaxis [33]. Since most telangiectatic lesions are found at the gastric and duodenal level, the first test to be performed is an upper endoscopy. If the latter is negative, the study can be extended with capsule endoscopy or enteroscopy. HHT patients with Smad4 mutation, a colonoscopy must be offered since its association with juvenile polyposis [49].

## 7. Treatment

Although HHT is one of the most prevalent rare diseases, there is currently no definitive treatment. However, more and more therapeutic strategies are being implemented to improve and control the symptoms derived from this disease. Table 3 summarizes the different therapeutic strategies used to address the clinical manifestations and complications of HHT [37].

### 7.1. Management of Epistaxis

#### 7.1.1. Care of the Nasal Mucosa

As a first line for epistaxis, the use of local hydrating therapies that moisturize the nasal mucosa is recommended, thus preventing bleeding. The use of saline and humidification has been shown to reduce the severity of epistaxis by avoiding erosions in the nasal mucosa [50]. Recent evidence shows no superiority between topical therapy (including tranexamic acid, beta blockers or estrogens) compared to moisturizing for preventing epistaxis [50]. In patients who persist with bleeding despite moistening agents, it is recommended to start with oral tranexamic acid [50,51,52,53]. Other systemic therapies including beta blockers or estrogens could be used on an individualized basis. Randomized studies have found promising results with tacrolimus and thalidomide [54,55,56].

#### 7.1.2. Invasive Treatments

As for ablative therapies, laser treatment, sclerotherapy, electrosurgery, and radiofrequency are available for nasal telangiectasias. Of these, laser treatment and sclerotherapy are the ones with the better cost-effectiveness [57,58,59]. Laser intervention has shown benefit in the treatment of epistaxis with no improvement in bleeding frequency after topical treatment [57,58,59]. Sclerotherapy, a more recent technique, is based on a local submucosal and subperichondrial injection of polidocanol, which causes vessel occlusion. This approach produces fewer complications than laser treatments, can be carried out with local anesthesia and as an outpatient setting, and prospective studies support its benefit in terms of reducing epistaxis [34,59]. Sclerotherapy requires expertise and is not a procedure that is readily available in most areas; it is only carried out in a few centers around the United States and Europe [60].

Other techniques used are radiofrequency and electrosurgery. For radiofrequency, a camera is used to enter the nasal turbinates, and through radiofrequency, the telangiectasias are ablated. Electrosurgery is a less commonly used technique, but a prospective study comparing it with laser treatment could suggest a superiority in terms of improved quality of life and a decrease in the frequency of bleeding [59]. Common risks of all these techniques are septal perforation, increased crusting, decreased airflow through the nares, loss of smell, and development of atrophic rhinitis [61].

These treatments are used in refractory cases of epistaxis in which less invasive local measures do not improve the patients’ quality of life. Today they are used less frequently as the new pharmacological therapy continues to evolve. Moreover, they have not shown superiority, and are reserved for cases in which the patient’s thrombotic risk is very high [36]. Septodermoplasty consists of replacing the mucosa of the anterior nasal cavity with a skin graft from the thigh. Harvey RJ et al. propose septodermoplasty in patients who have already undergone more than three laser ablation procedures and in whom another is contemplated in the next 6 months for increased epistaxis [62]. Young’s procedure or surgical closure of the nostrils is used in cases of hemodynamically compromised epistaxis or in patients who are dependent on blood transfusions despite optimal pharmacological therapy. Its recovery rate has been found to be better than septodermoplasty due to its mechanism of closing the airflow passage, thus completely stopping epistaxis [63]. Complications of these techniques include partial reopening or dehiscence and worsening sinus infections. In addition, about 25% of patients will need to continue with other small procedures despite the initial surgery. It is noted that the frequency of these techniques has decreased with the advent of medical therapies [34,62].

### 7.2. Approach to Pulmonary AVMs

Embolization is primarily conducted to prevent strokes and cerebral abscess. Treatment of pulmonary AVMs by embolization is recommended when the afferent artery of the malformation is at least 1 to 3 mm depending the expertise of the radiologist [34,42]. A high percentage of these malformations can be recanalized (18%), so it is recommended that these lesions be followed up and embolized if necessary. On the other hand, it is important that patients with diagnosed pulmonary AVMs use antibiotic prophylaxis prior to any dental intervention. In refractory cases or diffuse pulmonary AVMs, lung transplantation can be considered as a therapeutic option [46,64,65]. Patients with pulmonary AVMs are recommended to use filtered IVs and to avoid scuba diving [34,35].

### 7.3. Approach to Gastrointestinal AVMs

The technique of choice is endoscopic argon plasma coagulation of digestive AVMs. However, repeated argon plasma coagulation for recurrent AVMs in the GI tract, as a main treatment strategy, is not recommended [34]. Therefore, oral antifibrinolytics are also recommended, and in particular, tranexamic acid, which has been shown to reduce the number of endoscopic studies in these patients due to suspected gastrointestinal bleeding [66]. As in the treatment of epistaxis, if these patients persist with anemia despite previous treatment, initiation of systemic bevacizumab is recommended [34,35]. As in vascular ectasia on the intestinal wall bleeding, octreotide has been applied to HHT with some efficacy proven [67]. Other pharmacological treatments have also been used in these severe cases with less evidence. The use of other anti-angiogenic drugs, including estrogen receptor modulators such as tamoxifen, raloxifene, or bazedoxifene, is described in some cases. Other therapies are under investigation and include pazopanib, pomalidomide, and doxycycline. Evidence for most of these drugs is scant and larger trials are currently ongoing that may help elucidate the safety and efficacy of these drugs in HHT related bleeding [68,69,70,71,72].

### 7.4. Management of Central Nervous System AVMs

There is clinical uncertainty about how to manage these complications. Most decisions require multidisciplinary consent and will depend on the experience and recommendations of the local HHT center. Different techniques (surgical resection, interventional embolization, and radiosurgical ablation) can be performed depending on the size and location of the lesions. Other studies suggest following up with imaging tests in asymptomatic patients as long as they do not present changes in size or a high risk of rupture [73]. There is little evidence on which is the best option, so the decision should often be made by a multidisciplinary committee and the patient should be involved in the decision making process [34,41].

### 7.5. Pharmacological Treatment

#### 7.5.1. Antifibrinolytic Strategy

Antifibrinolytics are placed in the first line of treatment in HHT, aminocaproic acid and tranexamic acid the principal ones. The role of local tranexamic acid in the management of epistaxis has conflicting data results, but its systemic use has clearly showed benefits [51,52]. Small prospective trials [51,52,53] show that its systemic use decreases the frequency of epistaxis and improves hemoglobin levels. However, it should be noted that the lack of studies with a larger sample size makes it difficult to evaluate the potential risks of this treatment, especially the potential risk of producing thrombosis [37].

#### 7.5.2. Drugs That Stimulate ENG and ACVRL1 Gene Expression

Another therapeutic approach used has been the induction of *ENG* and *ACVRL1* gene transcription, with specific estrogen receptor modulating drugs (SERMs) being the most important. In women with osteoporosis, clinical trials with these drugs have been shown to improve the degree of osteoporosis and decrease the frequency of epistaxis. These drugs may be useful in HRT as they bind to the estrogen receptor and stimulate the Sp1 promoter, which activates transcription of the *ENG* and *ACVRL1* genes, compensating for functional haploinsufficiency [59].

Clinical trials with tacrolimus in patients with HHT and liver transplantation have shown an improvement in the control of epistaxis. This drug increases the expression of *ENG* and *ACVRL1*, stimulating tubulogenesis and cell migration [61].

Other antiangiogenic drugs used are non-selective beta-blockers such as propranolol and timolol; although their usefulness is mainly local, there are some clinical trials underway on their systemic use [64]. Within the line of angiogenesis inhibition, etamsylate has recently been approved. A drug that blocks the assembly of the complex that triggers (FGF) signaling at the cellular level. Pathway responsible for angiogenesis and closely related to HHT. The intervention study performed with this medication in Spain had certain limitations such as the number of patients and the short follow-up time; however, the results were promising and research with this drug should continue [74].

#### 7.5.3. Angiogenesis Inhibitor Drugs

Bevacizumab and thalidomide are anti-angiogenic drugs that act by blocking the VEGF (Vascular Endothelial Growth Factor) receptor. Bevacizumab has been shown to reduce the frequency of epistaxis, improve the degree of anemia, reduce the need for transfusions, reduce the progression of liver disease by delaying the need for transplantation, reduce the incidence of HOHF and improve quality of life. At present, these drugs are only indicated in patients with severe epistaxis, gastrointestinal bleeding or a combination of the two, as well as in patients with HOHF secondary to hepatic AVMs [65]. 

Phase II trials showed positive results in ESS and quality of life in patients receiving thalidomide as a treatment. Thalidomide treatment in HHT patients was associated with less red packed blood transfusion, decreased frequency of iron deficiency anemia, lower scores on the ESS, and thus improving the quality of life. Recent evidence shows that thalidomide makes blood vessels of HHT patients firmer and less prone to breaking [54,55].

#### 7.5.4. Potential New Treatments in HHT

One of the central avenues in the investigation of new drugs is the VEGF pathway, where bevacizumab is already in use. Nintedanib, a tyrosine kinase inhibitor, acts on the platelet-derived growth factor, the fibroblast growth factor, and the VEGF receptors. This drug is being investigated and has already been used in patients with HHT2 and pulmonary interstitial fibrosis. Sorafenib, a multikinase inhibitor, has shown to decrease gastrointestinal bleeding and prevent telangiectatic bleeding in different clinical trials [37,66,70]. Pazopanibshowed signs of efficacy in a small series [71]. Other trials with pomalidomide and doxycycline are ongoing [72].

Another pathway studied is that of the angiopoietin-2 receptor (ANGPT2), which also interacts with VEGF as a pro-angiogenic factor. Clinical trials have been initiated on anti-ANGPT2 antibodies that have shown normalization of vessel caliber in these patients [37].

The phase I trial INSIGHT is underway.to search the potential risks and usefulness of VAD044, an allosteric AKT inhibitor [75].

## 8. Conclusions

HHT is a complex congenital disease that involves different pathways and affects different organs. Its main symptomatology is associated with dysregulation of angiogenesis, translating as main clinical manifestations as epistaxis and telangiectasias. In the long term, several complications have been described such as recurrent infections, brain abscesses or AVMs in internal organs, pulmonary hypertension, and high output heart failure. Its clinical diagnosis is based on the Curaçao criteria but requires the development of the main signs and symptoms of the disease. However, diagnosis can now be made by genetic study, detecting pathogenic variants in the main genes related to HHT without the need to have developed the main symptoms of the disease. The complexity of its pathophysiology makes the therapeutic approach complex, so there is no definitive treatment for HHT, and the current approach is based on the treatment of its complications. The drugs used act at different levels of the molecular pathways of angiogenesis and are used to delay the development of complications.

## Figures and Tables

**Figure 1 jcm-11-05245-f001:**
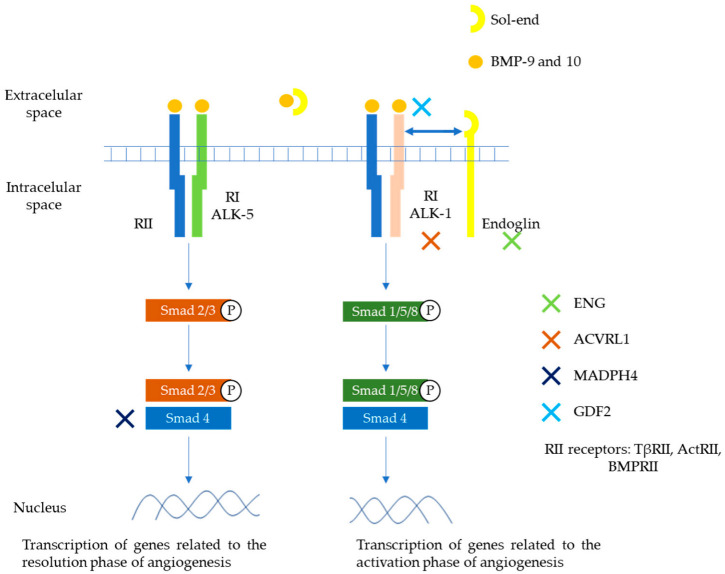
Main pathways involved in the pathophysiology of HHT. Members of the TGF-β family (mainly BMP9 and 10), bind to specific cell surface type I (R-I) and type II (R-II) receptors. These receptors exhibit serine/threonine kinase activity. The main R-I receptors involved in this pathway are ALK-1 and ALK-5. On the other hand, R-II receptors include TβRII, ActRII, BMPRII. Endoglin is an auxiliary receptor that associates with the ligand, ALK-1/R-II complex, potentiating its action. A soluble form of endoglin (sol-eng) can be generated by proteolysis of the membrane-bound receptor that can sequester ligands (BMP9 or 10) and thus modulate their binding to R-I/R-II receptors. The association between R-I (either ALK-1 or ALK-5) with R-II determines the specificity of ligand signaling. Upon ligand binding, R-II transphosphorylates R-I, which then propagates the signal by phosphorylating the receptor-regulated Smad family of proteins. Once phosphorylated, R-Smads form heteromeric complexes with a cooperating homologue called Co-Smad (Smad4) and translocate to the nucleus where they regulate the transcriptional activity of target genes. In endothelial cells, ALK1 and ALK5 activate signaling two different pathways via Smad1, 5, 8 (ALK1) or Smad2, 3 (ALK5), respectively. The first one promotes transcription of genes related to angiogenesis. The second promotes transcription of genes related to repairing phase. Endoglin, ALK1, Smad4, and BMP9 proteins are encoded by ENG, ACVRL1, MADH4, and GDF2 genes respectively. ActR, activin receptor; BMP, bone morphogenetic protein; BMPR, BMP receptor [14,15,16].

**Figure 2 jcm-11-05245-f002:**
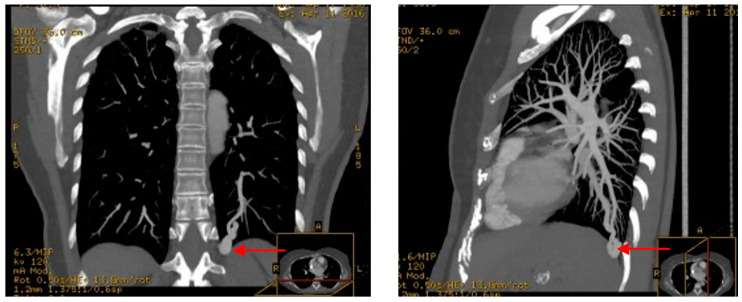
Pulmonary AVMs. AngioCT reconstruction of the thorax of a patient with HHT from the systemic and minority disease unit of the Hospital Ramón y Cajal, with the patient’s consent. Note the AVM located in the left lower lobe (red arrows).

**Table 1 jcm-11-05245-t001:** Genes responsible for HHT [6,7].

Gen	Affected Protein	Location	Phenotype	Frequency
*ENG*	Endoglin	9q34.11	HHT1	39–59%
*ACVRL1*	ALK1	12q13.13	HHT2	25–57%
*MADH4*	Smad4	18q21.1	HHT-juvenile polyposis syndrome	1–2%
*GDF2*	BMP9	10q11.22	HHT-like	<1%
*RASA-1*	p120-RasGAP	5q14.3	RASA-1 related disorders (CM-AVM)	Unknown

HHT, hereditary hemorrhagic telangiectasia; *GDF*, growth differentiation factor; p120-RasGAP, P120-Ras GTPase activating protein; CM–AVM, capillary malformation–arteriovenous malformation syndrome.

**Table 2 jcm-11-05245-t002:** Summary of clinical manifestations and their repercussions in patients with HHT [34,36].

Clinical Manifestations	Prevalence (%)	Comments
Epistaxis	90–95	Most limiting symptom for patients.
Telangiectasias	95	It can produce recurrent bleeding in the bearing areas of the body or those in contact with external surfaces such as the fingertips.
Anemia	50	It is associated with asthenia and chronic fatigue.
Pulmonary AVMs	15–50	Chronic hypoxaemia is only present in case of large pulmonary AVMs. Prevalence 10–20% in HT-II, 60% in HHT-I.
Hepatic AVMs	47–74	Three different types. Depending on their predominance, they increase the risk of HOHF, portal hypertension, hepatic encephalopathy, biliary ischemia, mesenteric ischemia, and hepatic cirrhosis.
Cerebral AVMs	2–20	Nonspecific symptoms (headaches or seizures)
Digestive AVMs	13–30	AVMs predominate in the stomach and duodenum.
Pulmonary hypertension	1–5	Can be caused by different mechanisms including hereditary group 1 PAH, or due to high cardiac output in the setting of liver AVMs (mostly associated with ACVRL1 mutation).

AVMs, arteriovenous malformation; HOHF, high output heart failure; PAH, pulmonary arterial hypertension.

**Table 3 jcm-11-05245-t003:** Therapeutic approach to HHT according to the different clinical manifestations that may occur [34,37].

Manifestations	Treatment	Comments
Epistaxis	Moisturizing (Topical hydration) Use of oral tranexamic acid if continuous bleeding despite moisturizing topical therapies	Recent evidence shows no superiority on all topical therapy (tranexamic acid, estrogens, propranolol, and bevacizumab) compared to placebo.
	Ablative treatments (laser, radiofrequency ablation, electrosurgery and sclerotherapy). Systemic therapy could be considered: beta blockers, thalidomide, tacrolimus. Antiangiogenics (Bevacizumab)	Consider if epistaxis persists despite topical treatments. Consider systemic therapy before surgery.
	Septodermoplasty	Consider in patients who do not respond to previous treatments.
Digestive bleeding	Endoscopic procedures are diagnostic and therapeutic. Consider capsule endoscopy if endoscopic bleeding is not identified.	Repeat sessions are discouraged to avoid repeated iatrogenic injury to the intestinal mucosa.
	In mild cases, oral antifibrinolytics may be considered.	
	If, despite previous treatments, bleeding persists, anemia requiring transfusions, antiangiogenic drugs (bevacizumab) can be initiated.	
Anemia	Oral ironIV iron if intolerant or lack of response to oral iron. Red blood cell transfusion.	A usual dose of 35 mg elemental iron tablets daily indicated.
Pulmonary AVMs	Transcatheter embolization: Consider in any AVM with afferent vessel >2 mm in diameter.	
	Chest CT is recommended to identify possible recanalization.	Follow-up with CT scan after embolization every 6 months, then every 3–5 years.
Cerebral abscess	If TTCE identifies the presence of a short circuit (although pulmonary AVM is not identified in CT): Antibiotic prophylaxis is recommended prior to dental procedures. Avoid administering air bubbles when cannulating veins.	
Pulmonary hypertension	Extend study to identify primary cause and address management (multidisciplinary consultation).	
Hepatic VMs	Most patients with symptomatic hepatic AVMs can be managed with medical treatment. Consider bevacizumab for patients who fail medical treatment.	
	Refer to referral center to consider liver transplantation in patients with refractory symptomatic hepatic AVMs (HOHF, biliary ischemia, or complicated portal hypertension).	Liver biopsies should be avoided in patients with HHT.
Cerebral AVMs	Treated depending on risk of bleeding and expertise of the neurosurgical team. Embolization or stereotactic radiosurgery depending on the size, location, and symptomatology.	

AVMs, arteriovenous malformation; VM, vascular malformation; HOHF, high output heart failure; TTCE, transthoracic contrast echocardiography; CT, contrast tomography.

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
