# Peer review of "Hereditary Hemorrhagic Telangiectasia: Genetics, Pathophysiology, Diagnosis, and Management"

_jcm, 2022, doi:10.3390/jcm11175245_

Round 1

Reviewer 1 Report

Dear Authors,

first of all, I wish to thank you for giving me the opportunity to read this your interesting review.

I have only this minor comment: the text is not always easy to read because several phrases are too long and contorted. I suggest to re-edite the manuscript, writing shorter sentences.

In addition, some typos should be removed. By way of example: bone morrow (page 4) or bone marrow ?  sun exposure (page 3, line 93) and not Sun...; page 3, line 91: please, modify; 5. Pulmonary hypertension and HHT in page 6 (there is already a paragraph no. 4) .....and so on (please read the pages no. 7 and no.8).  

Reviewer 2 Report

I have carefully read the review entitled “Hereditary Hemorrhagic Telangiectasia: Genetics, pathophysiology, diagnosis, and management”. It is a very comprehensive and well-written article. However, there are some minor issues I want to underline and mention

1 There are some studies which I think undervalued in HHT. These studies report a good example of genes and clinical manifestations. In my opinion these studies are worth deserved to be mentioned and cited in this review.

10.4274/balkanmedj.galenos.2019.2019.7.2

https://doi.org/10.1007/s10038-007-0187-5

10.1007/s00439-004-1196-5

10.1080/03009734.2018.1483452

2.The data regarding the use of thalidomide is lacking. There is phase 2 study and retrospective analysis which deserved to be cited and mentioned

10.1016/S2352-3026(15)00195-7

10.4274/tjh.galenos.2018.2018.0190

Reviewer 3 Report

The authors write a nice review on HHT genetics, pathophysiology, diagnosis, and management. 

Minor edits: 

-line 52/53: grammar is odd. Would write instead: All affected genes encode proteins that participate in the TGFb signaling pathway. 

-In table 1: would say GDF2 is HHT-like (under phenotype). Also, the unknown gene after the GDF2 is the RASA 1 gene, which I would describe under phenotype as RASA-1 related disorders (CM-AVM). I would take out the last unknown from that table. 

-line 61, generate instead of generated

-line 69: typo- HHT instead of THH

line 119-120: revise grammar.

line 148-149: What do you mean by urgent situations? Regarding targets for anemia, would describe both targeting hemoglobin and ferritin levels. 

Table 2: I don't understand what you mean by produce recurrent bleeding in support areas. Please clarify. Digestive avm prevalence seems low. Can you check multiple sources to confirm? Under pulmonary HTN comments-would say under comments: Can be caused by different mechanisms including hereditary group 1 PAH, or due to high cardiac output in the setting of liver avms (mostly associated with ACVRL1 mutation). 

Line 160/161: I would state that most pulmonary avms are asymptomatic, but can rarely cause hypoxemia and/or digital clubbing. Other rare complications include hemoptysis or hemothorax. 

Line 181-181: Would change to: The most frequent clinical manifestations of liver avms are symptoms related to high cardiac output heart failure, including dyspnea and edema. Other patients can present with abdominal pain from biliary ischemia or anicteric cholestasis. 

Line 190- please specify which genotype

Line 194-seizures (instead of epilepsies)

Line 220-normal or low cardiac output.

Line 244-247: In all patients with HHT, an echo with bubble study is recommended every 5 years. Only if bubble study is positive, should a CT scan be done. In patients who continue to have positive bubble studies despite pulmonary avm embolization, a CT scan can be done every 5 years. 

Line 249-250: Most HHT experts recommend brain imaging (1 MRI with and without contrast, and repeat as an adult if first one done as a child) for screening. I'm not sure I would say that this is debatable in adults, but you can say that it is somewhat controversial due to some data suggesting low risk of complications. 

Line 253-254: "the study should be extended", should be changed to: additional imaging can be considered with the use of MRI...

Table 3: I think this table is messy and includes a lot of treatments that are not approved for HHT or have been shown to be ineffective in clinical trials. This should be noted somewhere (for instance, all the topical treatments were negative in the NOSE trial). I would revise the epistaxis and GI bleeding section. Would take out the comment about oral beta blockers, tacrolimus, and octreotide could be considered. There is no data to support this statement.  Under cerebral abscess: avoid administering air bubbles when cannulating veins (not airways). Would refer to the new HHT guidelines to improve this table and the recommendations (hhtguidelines.org). 

Line 290: would mention that sclerotherapy requires expertise and is not a procedure that is readily available in most areas. Only done in a few centers around the US. 

Line 331: would talk about the complications/morbidity of septal dermoplasty and Youngs procedure, and how with the advent of medical therapies, these procedures have mostly fallen out of favor. 

Line 315-318: Mention that embolization is done primarily to prevent strokes and cerebral abscess. Also would mention that patients with pulmonary avms are recommended to use filtered IVs and to avoid scuba diving. 

Lines 328-330: Would emphasize that the data for most of these drugs is scant and that larger trials are currently ongoing (pomalidomide, pazopanib, and doxy) that may help elucidate the safety and efficacy of these drugs in HHT related bleeding. Plasma coagulation is only recommended as a temporizing measure, until a patient can get better control with medical therapy. Repeated argon plasma coagulation for recurrent avms in the GI tract, as a main treatment strategy, is not recommended. 

Line 332: not sure surgical resection is recommended over embolization or ablation. There is clinical equipoise here and lack of data comparing these different strategies. One should rely on the expertise and recommendation of their local HHT center. 
